

# Detection of offensive terms in resource-poor language using machine learning algorithms

Muhammad Owais Raza[1], Naeem Ahmed Mahoto[1], Mohammed Hamdi[2], Mana Saleh Al Reshan[3], Adel Rajab[2] and Asadullah Shaikh[3]

[1] Department of Software Engineering, Mehran University of Engineering and Technology Jamshoro, Jamshoro, Pakistan
[2] Department of Computer Science, Najran University, Najran, Najran, Saudi Arabia
[3] Department of Information Systems, Najran University, Najran, Najran, Saudi Arabia

Corresponding author
Mohammed Hamdi,
mahamdi@nu.edu.sa

## ABSTRACT

The use of offensive terms in user-generated content on different social media platforms is one of the major concerns for these platforms. The offensive terms have a negative impact on individuals, which may lead towards the degradation of societal and civilized manners. The immense amount of content generated at a higher speed makes it humanly impossible to categorise and detect offensive terms. Besides, it is an open challenge for natural language processing (NLP) to detect such terminologies automatically. Substantial efforts are made for high-resource languages such as English. However, it becomes more challenging when dealing with resource-poor languages such as Urdu. Because of the lack of standard datasets and pre-processing tools for automatic offensive terms detection. This paper introduces a combinatorial pre-processing approach in developing a classification model for cross-platform (Twitter and YouTube) use. The approach uses datasets from two different platforms (Twitter and YouTube) the training and testing the model, which is trained to apply decision tree, random forest and naive Bayes algorithms. The proposed combinatorial pre-processing approach is applied to check how machine learning models behave with different combinations of standard pre-processing techniques for low-resource language in the cross-platform setting. The experimental results represent the effectiveness of the machine learning model over different subsets of traditional pre-processing approaches in building a classification model for automatic offensive terms detection for a low resource language, *i.e.*, Urdu, in the cross-platform scenario. In the experiments, when dataset D1 is used for training and D2 is applied for testing, the pre-processing approach named Stopword removal produced better results with an accuracy of 83.27%. Whilst, in this case, when dataset D2 is used for training and D1 is applied for testing, stopword removal and punctuation removal were observed as a better preprocessing approach with an accuracy of 74.54%. The combinatorial approach proposed in this paper outperformed the benchmark for the considered datasets using classical as well as ensemble machine learning with an accuracy of 82.9% and 97.2% for dataset D1 and D2, respectively.

# INTRODUCTION

The advancements in information and communication technologies have decreased barriers and have flooded immense amounts of user-generated content. Social media, including Facebook, Twitter and other such platforms, have offered discourse and raised viewpoints about numerous topics and stories around the world (*Peters et al., 2022*). The caveat is the misuse and provocation of hate and offence. The trend in cyberbullying and online harassment is on the rise due to the liberty of social media platforms (*Masadeh, Davanager & Muaad, 2022*). The offence, harassment and cyberbullying are grave concerns on social networking platforms. The challenge is the detection of such contemplative views in user-generated content. Manually, it is next to impossible to recognize and filter out offensive comments in online discussions. Machine learning (ML) and natural language processing (NLP) are the evidence to cope with such challenges in the existing literature. Several approaches have been employed to detect offensive and hate terms in online text, such as lexicon-based approach (*Gitari et al., 2015*), n-gram approach (*Ptaszynski et al., 2019*; *Sigurbergsson & Derczynski, 2019*), and ensemble learning (*Pelle, Alcântara & Moreira, 2018*).

The methods for detecting offensive language for resource-rich languages (*e.g.*, English) have been proposed in abundance, as shown in this section. Although multilingual content is available on social media platforms, yet little efforts have been made to cater for the resource-poor languages (*Cunliffe et al., 2022*). This has increased the need for automatic offensive language detection systems for low or poor-resource languages. There are some research studies for the languages such as German (*Schneider et al., 2018*), Danish (*Sigurbergsson & Derczynski, 2019*), and Arabic (*Alakrot, Murray & Nikolov, 2018*). These studies about low-resource language have used ML approaches with a single dataset for both training and testing. The success of the classification model for low-resource languages in this world of numerous social media sites relies on available resources such as datasets and pre-processing tools and its evaluation of cross-platform data to reduce generalization error.

This study proposes combinatorial pre-processing techniques along with cross-platform datasets for the development of a classification model using classical machine learning methods. Deep learning and transfer learning are beyond the scope of this study, and thus, these learning approaches are not applied. The major contributions of this study are:

1. Highlighting and discussing the impact of standard pre-processing approaches over cross-platform abusive language datasets.
2. Evaluation of pre-processing approach over the performance on classifier for offensive language detection for cross-platform datasets.
3. Beating state-of-the-art results over benchmarked dataset for YouTube comments using a combination of standard pro-processing steps.

The rest of the paper is structured as follows: the 'Literature Review' section reports a literature review and relevant studies. The 'Methodology' section shows the methodology employed in this study. The 'Experimental Results' section discusses the experimental results and finally, conclusions are drawn in the 'Conclusions and Future work' section.

# LITERATURE REVIEW

Recently, the attention of computational linguistics is growing to analyse the immoral conduct of individuals on social networking sites like Facebook (*Gitari et al., 2015*), and Twitter (*Schneider et al., 2018*). Groups and individuals from all over the globe post messages on social networking sites and receive dozens of negative or positive responses or comments on the posted things. These remarks generally include derogatory terms or harsh language due to the sheer differences among individuals of different ethnicity, religion, culture, or nations (*Burnap & Williams, 2015*). These abusive or despised statements initiate cyberbullying

This existing literature in the field of abusive language detection and profanity detection includes (*Agrawal & Awekar, 2018*; *Nobata et al., 2016*; *Malmasi & Zampieri, 2018*). In *Ptaszynski, Eronen & Masui (2017)* researchers use different classification methods for cyberbullying detection, including support vector machines, naive Bayes, k-nearest neighbours, J48, JRip, random forest, and convolutional neural network (CNN), and the results indicate that CNN outperforms the other classifiers by more than 11 percent in F-score. *Ibrahim, Torki & El-Makky (2018)* suggests a combination of three models: CNN, bidirectional long short-term memory (BLSTM), and bidirectional gated recurrent unit neural network (BGRU). The suggested method separates prediction into two parts and outperforms existing techniques in terms of the F1-score. Because the dataset is significantly unbalanced, several data augmentation approaches are applied to address the class imbalance issue. *Van Aken et al. (2018)* compares several DL methods and shallow learning methods and presents an ensemble model that surpasses all individual models. Individual models include BLSTM, long short-term memory (LSTM), CNN, BGRU with attention, logistic regression, and the word embeddings utilised were based on Fastext and Glove. The most common feature selection techniques are bag of the word (*Watanabe, Bouazizi & Ohtsuki, 2018*), and n-grams (*Rani & Ojha, 2019*).

Most of the available literature on this use case primarily focuses on English because of the huge availability of NLP resources for English such as big annotated datasets and NLP toolkits. But for resource-poor languages such as Urdu very limited resources are available. This limits the research in this area but also makes it an ideal resource-poor language to tackle the problem of abusive language detection. As in *Akhter et al. (2021)* researchers have used machine learning and deep learning techniques to understand which technique performed well on Roman Urdu and Nastaliq Urdu scripts. In order to improve collaboration and contribution in the field of Urdu abusive language detection, a competition was arranged to come up with novel ways to detect the abusive language in Urdu Nastaliq script (*Amjad et al., 2022*). Urdu, a language spoken mainly in Pakistan and India, is considered a low-resource language in terms of natural language processing (NLP) research. Urdu has been the focus of this study, especially, aiming at the specific challenges of detecting offensive terms in Urdu. It is an open challenge in Urdu due to the fact that it lacks comprehensive language resources and tools, such as annotated corpora, lexicons, and part-of-speech taggers. These resources are necessary for developing and evaluating

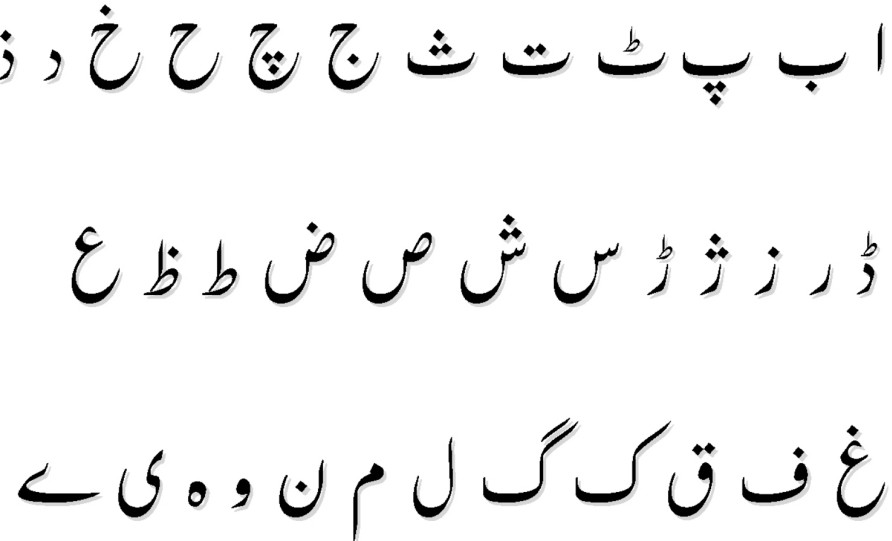

**Figure 1**   **Nastaliq Urdu alphabet.**

offensive language detection models. Besides, Urdu has a rich vocabulary and uses multiple scripts, which further complicates the task of detecting offensive terms in the language.

Urdu is available in two scripts one is Roman, and the other is Nastaliq. In this study, the Nastaliq-styled Urdu script is used. Nastaliq Urdu is morphologically very rich. Figure 1 shows the alphabet and structure of Nastaliq Urdu. It is right-to-left written language. It is context-sensitive but there is no capitalization. The Urdu language is an important research language in South Asia (*Daud, Khan & Che, 2017*) with more than 230 million speakers worldwide (*Amjad et al., 2022*). Due to complex morphology, grammatical restriction, and low availability of resources, automatic detection of abusive language detection is a layered and complex machine learning task. Because of the aforementioned problems, very limited work is done in this area and it is the reason for picking Urdu as a use case of this study.

Several machine learning techniques have been used to detect the offensive language in Urdu, as *Hussain, Malik & Masood (2022)* used embedding to train the classifier, *Humayoun (2022)* used feature combination, *Ali et al. (2022)* used transfer learning, and *Das, Banerjee & Mukherjee (2022)* used data bootstrapping Table 1 shows machine learning (ML) and deep learning (DL) techniques for classification, datasets used for the model, pre-processing technique and feature selection (*i.e.*, term frequency-inverse document frequency (TFIDF), n-gram, lexicon or word embeddings). All of these studies use linear pre-processing (a complete dataset is pre-processed at once using all adopted pre-processing steps linearly). The classification model is trained on the dataset from a single social media platform that restricts the usage and validity of the model. In this study, machine learning is used, where the model is trained and tested on cross-platform datasets using different machine learning algorithms. Unlike other studies, the focus of this study is not to come up with a novel ML solution for the detection of offensive terms in resource-poor language to the problem but to study the effects of existing resources

**Table 1  Research studies for resource-poor language—Urdu.**

| Reference | Language | Technique | Cross platform dataset | Pre-processing | Feature selection |
|---|---|---|---|---|---|
| Amjad et al. (2021) | Urdu | ML DL | No | Linear | TFIDF |
| Haq et al. (2020) | Urdu | ML | No | Linear | Manual Lexicon |
| Das, Banerjee & Saha (2021) | Urdu | ML DL | No | Linear | Word embeddings |
| Akhter et al. (2020) | Urdu | ML | No | Linear | N-Gram |
| Humayoun (2022) | Urdu | ML | No | Linear | TFIDF |

for pre-processing on a cross-platform data and answer that can these techniques be so efficient that these can outperform a SOTA results for the dataset using combinatorial pre-processing (pre-processing techniques where combinations of existing standard preprocessing are applied). The following research questions are developed based on the literature review performed in this study.

**RQ1:** What is the impact of combining different standard pre-processing steps over the performance of automatic detection of offensive terms in low-resource language? From Table 1, it can be seen that linear prep-processing approaches are being used in the literature. Thus, there is a need to study the impact of combining different preprocessing techniques.

**RQ2:** What were the most suitable combination of prep-processing techniques for effective results over those datasets? The 'Literature review' section of this study also reports studies in which ML techniques were considered suitable for certain cases. The reported studies raised a similar question for prep-processing techniques.

**RQ3:** How do the models in this study perform when compared to results from the models in the literature in *Akhter et al. (2020)*?

# METHODOLOGY

Figure 2 represents the methodology adopted in this research study. There are four main blocks: (i) data gathering, (ii) pre-processing, (iii) training & testing set selection and (iv) modelling and evaluation. The details of each block are discussed in detail in the following.

## Data gathering

For the purpose of this study, two cross-platform datasets have been gathered for detecting offensive language, one from Twitter platform (*Amjad et al., 2022*) (referred to as $D_1$) and the other from YouTube (referred to as $D_2$). The type of text on both platforms is different so these datasets are ideal for answering the research question discussed in the introduction.

The $D_1$ dataset comprised of 2,400 tweets containing offensive language in Urdu. The statistics of the datasets used in this study are reported in Fig. 3. The $D_1$ is a balanced dataset with 1,187 offensive tweets and 1,213 non-offensive tweets. Likewise, $D_2$ is also a balanced dataset with 1,109 offensive comments and 1,062 non-offensive comments.

A sample representation of datasets $D_1$ and $D_2$ are shown as word clouds in Figs. 4 and 5 respectively displaying the most prominent words. The word cloud is created using

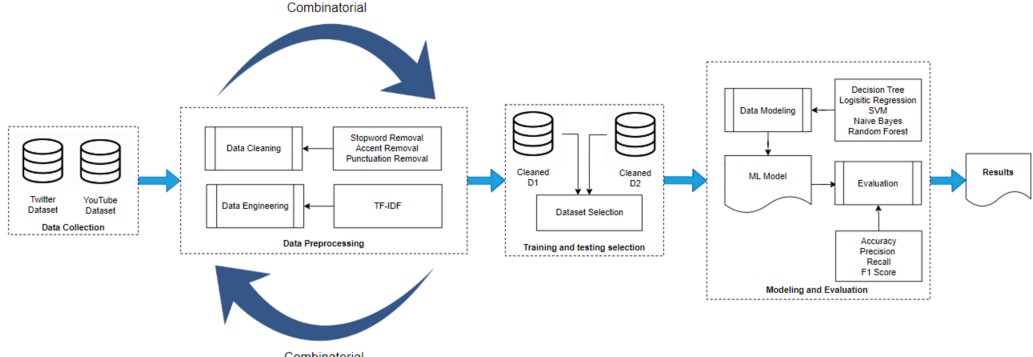

**Figure 2** Methodology diagram.

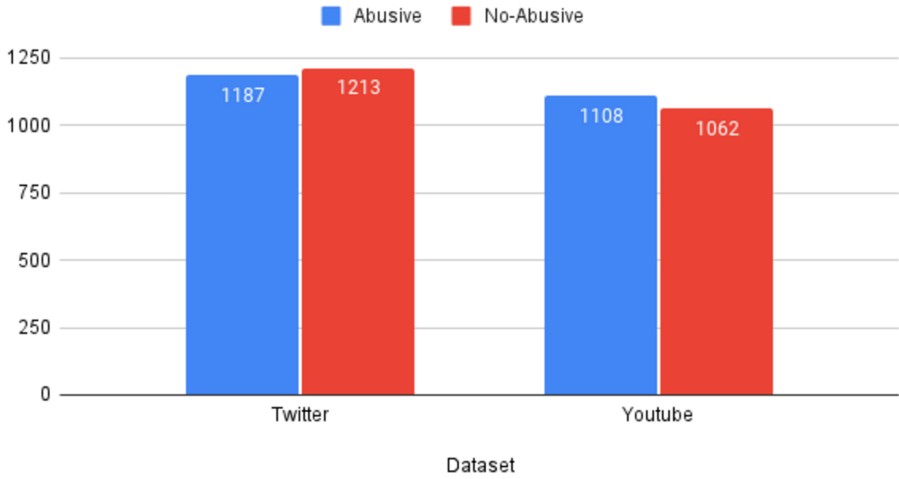

**Figure 3** Statistics of the datasets.

Matplotlib and WordCloud package in Python. The words in the word cloud are from text columns from both datasets. The parameters defined for the word cloud are the usage of available stopwords, size of fonts, number of most frequent words and colourmap to have a consistent colour theme for both word clouds.

The collected datasets are pre-processed in order to transform them into a suitable format for further processing. This study proposes a combinatorial pre-processing approach using standard pre-processing techniques. The details are described in the following subsections.

## Pre-processing

Data pre-processing steps in this study consist of two main operations: (i) data cleaning data and (ii) feature engineering. To clean data, stopword removal (SR), punctuation removal (PR) and accent removal (AR) are performed.

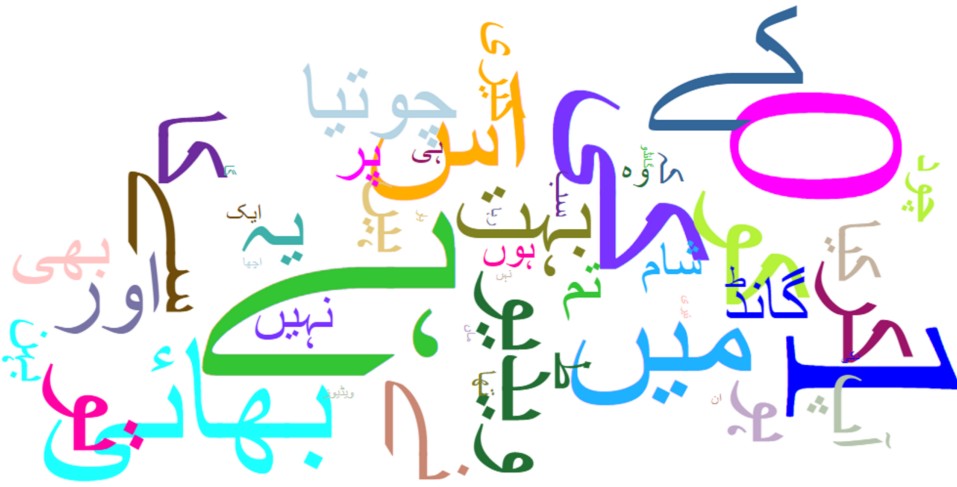

**Figure 4** Word cloud of $D_1$ dataset.

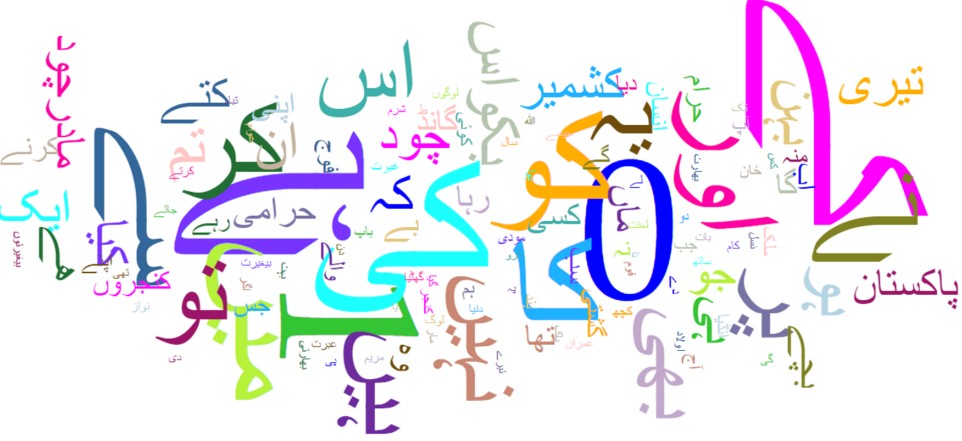

**Figure 5** Word cloud of $D_2$ dataset.

### Stopword removal (SR)

Stopword are usually in large quantities but have the least significance in the text classification. For instance, *is, are, am, was, were, the, a, an* etc are examples of stopwords in English and Fig. 6 shows the example of stopwords in Urdu. These words are removed in the process of stopword removal. An example of stopword removal is given in Fig. 7.

### Punctuation removal (PR)

Punctuation removal is the process of removing all the punctuation in a given sentence example of punctuation removal is given in Fig. 7.

| Sample Stopwords List | | | | | |
|---|---|---|---|---|---|
| Urdu | English | Urdu | English | Urdu | English |
| اور | And | آپ | You | تھا | Was |
| ابھی | Now | آتا | Come | جیسا | Like |
| اپنا | Our | پاس | Close | دے | Give |
| اگر | If | تاکہ | Because | ذرا | Part |
| ان | They | تم | You | رکھا | Put |

**Figure 6  Sample stopwords list for Urdu.**

| Urdu Text Preprocessing | |
|---|---|
| **Puntuation Removal** | |
| Urdu | English |
| کر سکتی ہے؟ | Can do? |
| **Before** | **After** |
| کر سکتی ہے؟ | کر سکتی ہے |
| **Stopword Removal** | |
| Urdu | English |
| توتتئر ۓ بکواس کرنا ہی آسان ہ | It's easy to talk nonsense on Twitter |
| **Before** | **After** |
| توتتئر ۓ بکواس کرنا ہی آسان ہ | توتتئر ۓ بکواس آسان |
| **Accent Removal** | |
| Urdu | English |
| دالتِ عظمیٰ درخواست | A Supreme Court application |
| **Before** | **After** |
| دالتِ عظمیٰ درخواست | دالت عظمی درخواست |

**Figure 7  Text cleaning inputs and results.**

### Accents removal (AR)

Accent removal is the process of removing accents from the accented Unicode characters in Urdu text; it can be done in two ways (1) by changing them into ASCII (2) by removing them completely. An example of accent removal is given in Fig. 7.

Urdu is among resource-poor languages as discussed above, which does not have a standard list of stopwords. Therefore, the list of stopwords in Urdu are created, which is available at GitHub (https://github.com/Delta-Sigma/urdu-stopwords/blob/master/urdu_stopwords.txt). The stopword list contains 265 commonly used words in the Urdu language, that have the least significance. Later, punctuation removal and accent removal operations are performed for cleaning the datasets. These two operations (*i.e.*, PR and AR) are performed using urduhack library (https://github.com/urduhack). The three operations

**Table 2 Recursive pre-possessing iterations.**

| Iteration # | Stopword removal (SR) | Punctuation removal (PR) | Accent removal (AR) | TF-IDF scheme | Dataset |
|---|---|---|---|---|---|
| 1 | ✓ | ✗ | ✗ | ✓ | $D_1$ |
| 2 | ✓ | ✓ | ✗ | ✓ | $D_1$ |
| 3 | ✓ | ✓ | ✓ | ✓ | $D_1$ |
| 4 | ✓ | ✗ | ✗ | ✓ | $D_2$ |
| 5 | ✓ | ✓ | ✗ | ✓ | $D_2$ |
| 6 | ✓ | ✓ | ✓ | ✓ | $D_2$ |

SR, PR and AR are performed stand-alone as well as in combination *Combinatorial Pre-processing* for the development of the classification model on cross-platform datasets.

The term frequency-inverse document frequency (TF-IDF) scheme is used as vector space representation (vectorization) model in this study. The reason behind using TF-IDF is the evidence in the literature that it performs better than the other vectorization schemes such as bag-of-words and n-grams. TF-IDF is a measure to calculate the significance of a word in a document and it is computed as follows:

$$W_{i,j} = tf_{i,j} * log \frac{N}{df_i} \tag{1}$$

where $W_{i,j}$ represents the significance of a word in document, $tf_{i,j}$ shows the number of times i occurs in j and $df_i$ shows number of documents which contains i. N is the total number of documents.

The combinatorial pre-processing approach is applied with multiple iterations. The number of feature vectors and labels used for modelling is the same as the number of iterations. Table 2 shows the iterations and combination of pre-processing techniques used in each iteration. Iteration 1 (record number 1 in Table 2) shows that the feature vector created is cleaned with stopword removal (SR) and uses TF-IDF scheme is applied for the dataset $D_1$. Similarly, iteration 2 shows that both SR and PR are applied in conjunction with the TF-IDF scheme for $D_1$. The 3rd iteration represents SR, PR, and AR cleaning techniques applied with TF-IDF feature engineering for $D_1$. Likewise, the combinatorial approach is also applied to the dataset $D_2$. Thus, Table 2 reports these six (06) iterations for both datasets. Each iteration produces a vector containing a maximum of 10,000 words in the vocabulary.

## Training testing sets

The considered cross-platform datasets $D_1$ and $D_2$ have been used for the training and testing using the split percentage method of validation. The study proposes the use of datasets from different platforms for training and dataset from entirely different platform testing, thus, the training set can be taken at different percentages from one dataset and tests can be performed on the entirety of the other dataset as shown in Table 3. The four experiments with 100%, 80%, 70% and 60% are taken as training sets from $D_1$. The developed model is used for testing over a $D_2$. Likewise, four experiments with 100%, 80%, 70% and 60% were taken as training sets from $D_2$ and tested on complete $D_1$.

**Table 3  Training testing sets scenarios.**

| Scenario | Experiments | Training (%) | Testing (%) |
|---|---|---|---|
| Scenario 1 | 1 | $D_1$ (100) | $D_2$ (100) |
| | 2 | $D_1$ (80) | $D_2$ (100) |
| | 3 | $D_1$ (70) | $D_2$ (100) |
| | 4 | $D_1$ (60) | $D_2$ (100) |
| Scenario 2 | 5 | $D_2$ (100) | $D_1$ (100) |
| | 6 | $D_2$ (80) | $D_1$ (100) |
| | 7 | $D_2$ (70) | $D_1$ (100) |
| | 8 | $D_2$ (60) | $D_1$ (100) |

## Modelling and evaluation

The machine learning algorithms used in the study are well established, namely, decision tree (DT), logistic regression (LR), support vector machine (SVM), naive Bayes (NB), and random forest (RF). The selection of machine learning algorithms in this study has been made carefully keeping several factors as the selection criteria such as the availability of data, the research questions, and the complexity of the problem. Besides, due to the small or limited number of instances in the considered datasets, these selected algorithms deem suitable for this study. The considered machine learning algorithms usually perform well when applied to small datasets. The details of the machine learning algorithms applied in the experimental set-up are described in the following.

## Decision tree

A decision tree (DT) is a general-purpose predictive modelling method with applications in a variety of fields. It is among the most commonly used and practical supervised learning methods. DT is a tree-like graph in which nodes represent the points at which a feature is selected and pose a question; edge denotes the answers to the question, and leaves show the actual class label. They are employed in nonlinear decision-making. The DT method has several advantages over other methods: (a) It is simple to interpret and understand. (b) It is simple to depict graphically. (c) It can handle both quantitative and qualitative data. (d) It necessitates minimal data preparation. (e) It handles large datasets well. Figure 8 shows a simple example of a decision tree employed for weather prediction (*Matzavela & Alepis, 2021*).

## Logistic regression

Logistic regression (LR) is also another effective supervised machine learning approach for binary classification. Logistic regression fits well as a linear regression for classification tasks. To predict a binary output variable, logistic regression employs the logistic function shown in Eq. (2). The major distinction between logistic regression and linear regression is the range of logistic regression, which is limited to 0 and 1. Furthermore, logistic regression unlike linear regression doesn't really require a linear connection between input and output variables. This is because the odds ratio is transformed using a nonlinear log transformation

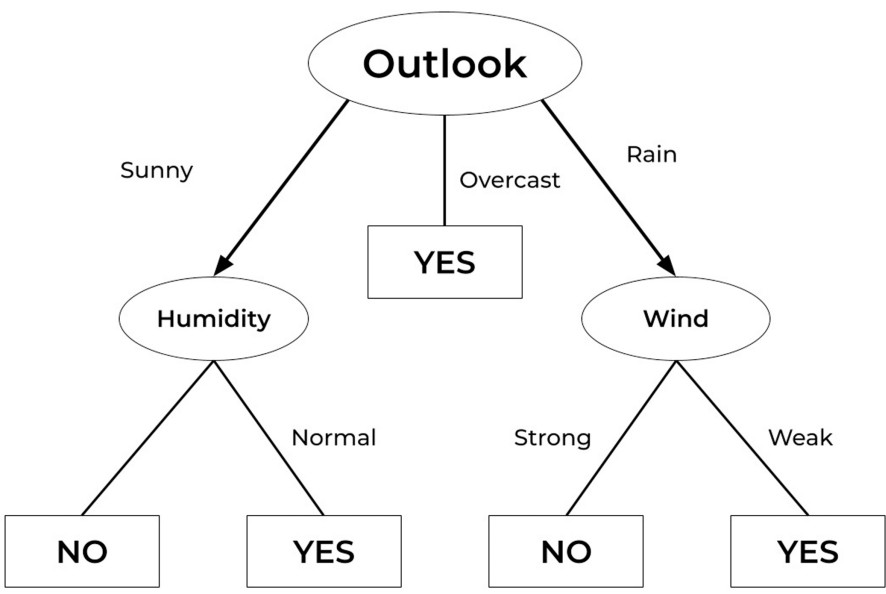

**Figure 8  Decision tree example.**

(*Belyadi & Haghighat, 2021*).

$$LR = \frac{1}{1 + e^x}. \tag{2}$$

The input variable in the sigmoid function Eq. (2) is x. Consider feed values ranging from −20 through 20 into logistic function as shown in Fig. 9. The inputs have been converted to values ranging from 0 to 1.

## Support vector machine

The support vector machine (SVM) is a prominent supervised learning technique that is used for both classification and regression problems. However, it is mostly utilised for classification tasks. The goal of the SVM algorithm is to find the optimum decision or line boundary for categorising n-dimensional space such that fresh data points be placed at the proper category in future. A hyperplane is the optimal choice boundary. SVM selects the extreme vectors/points that aid in the creation of the hyperplane. Consider Fig. 10 that shows two distinct categories separated by a hyperplane or decision boundary (*Mohammadi et al., 2021*).

## Naive Bayes

Bayesian classifiers are a type of classification method that is based on Bayes' Theorem. It is a group of algorithms that all share a similar proposition, namely that every pair of characteristics being categorised is independent of one another. Bayes' theorem, often called the Bayes rule, is a mathematical formula used to calculate the likelihood of a hypothesis given past knowledge. It is determined by the conditional probability (*Bhavani & Kumar,*

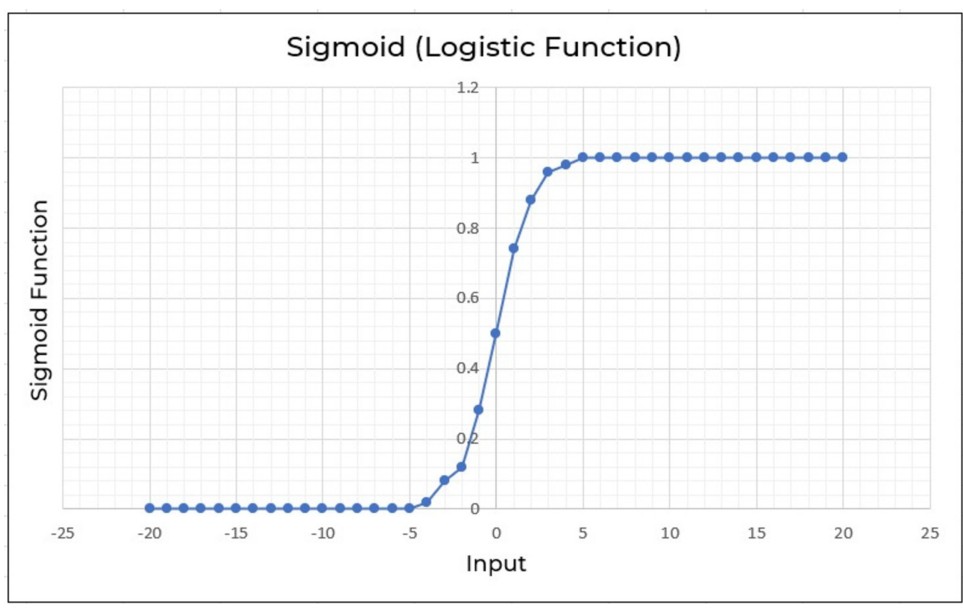

**Figure 9** Logistic regression employed to a range of −20 through 20.

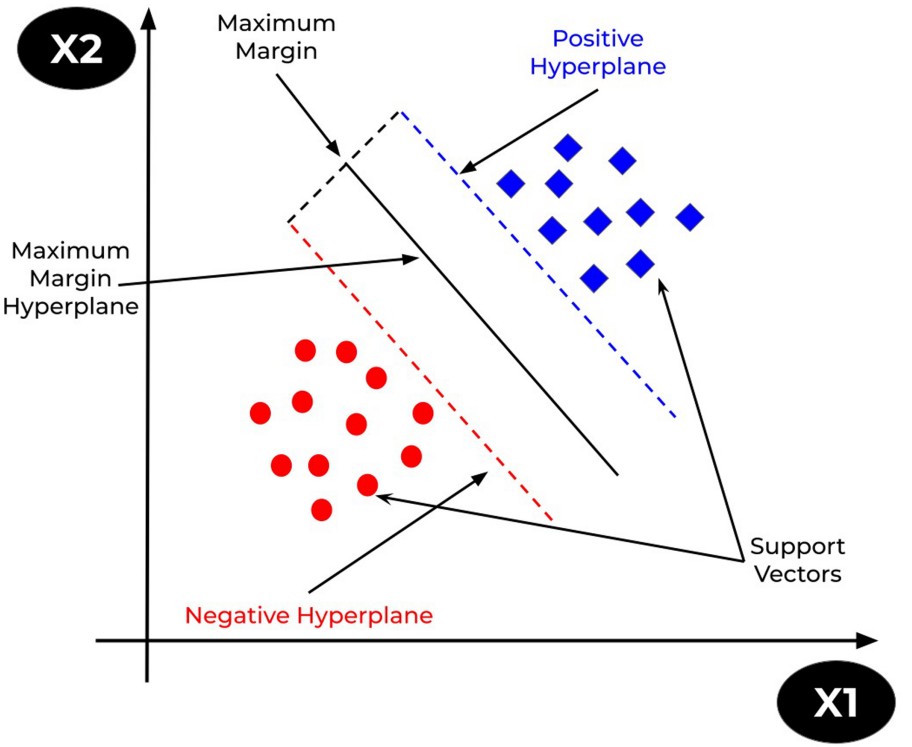

**Figure 10** SVM representation.

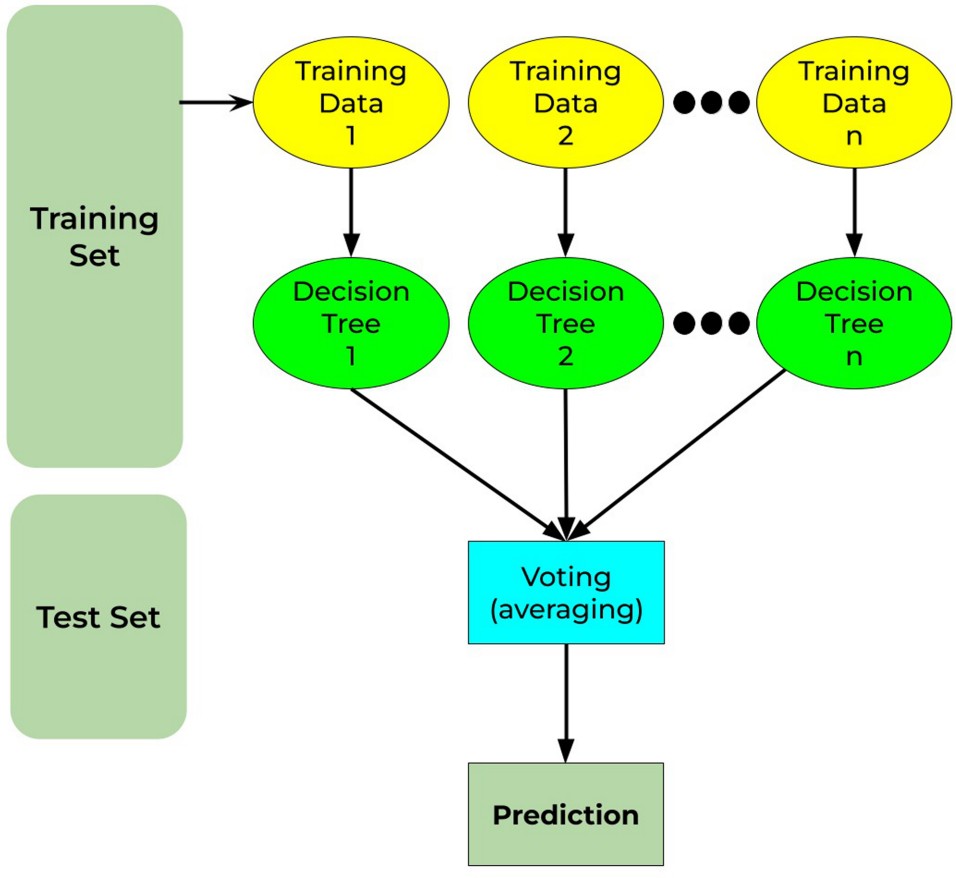

**Figure 11** **Random forest working principle.**

*2021*). The formula for the Bayes rule is shown in Eq. (3).

$$P(A|B) = \frac{P(B|A) * P(A)}{P(B)}. \tag{3}$$

## Random forest

Random forest is a well-known machine-learning algorithm. It may be applied to both regression and classification problems. It is built on the notion of ensemble methods, which is the process of merging numerous classifiers to resolve a complicated issue and enhance the model's performance. It comprises decision trees on different subsets of the provided dataset and chooses the average to enhance the predicted accuracy of that dataset (as the name implies—random forest). Instead of depending on a decision tree classifier, the random forest collects the forecasts from each tree and predicts the output depending on the voting of predictions (*Sumathi & Pugalendhi, 2021*). Figure 11 shows the working principle for the random forest as discussed.

After modelling the next step is to evaluate the performance of different machine learning algorithms with experimental set-up. The performance evaluation metrics used in this study

are accuracy, precision, recall and F-measure. Details of the considered evaluation metrics are described as follows.

### Accuracy

Accuracy metric of a model is shown as the ratio of correctly predicted text labels to the total number of instances in the dataset. Accuracy is computed as reported in Eq. (4).

$$Accuracy = \frac{T_p + T_n}{T_p + T_n + F_p + F_n} \tag{4}$$

where $T_p$, is true positive, Tn refers true negative, $F_p$ shows false positive and $F_n$ means false negative.

### Precision

An algorithm's precision is given as the ratio of correctly predicted value with the total number of predicted values. It is calculated as shown in Eq. (5).

$$Accuracy = \frac{T_p}{T_p + F_p}. \tag{5}$$

### Recall

Recall of a model is a metric that is represented as the ratio of correctly classified values divided by the total number of values in the dataset. It is computed by Eq. (6).

$$Accuracy = \frac{T_p}{T_p + F_n}. \tag{6}$$

### F-measure

F1 score also called F-measure is the harmonic mean of the values. It is considered a a reliable metric, which includes both precision and recall in computing its value. F1 score is harmonic or precision and recall. F1 score provides a balance between precision and recall. Further, F1 score relatively suits better when the distribution of class labels is imbalanced. It is computed by Eq. (7).

$$F - measure = \frac{2 * precision * recall}{precision + recall}. \tag{7}$$

## EXPERIMENTAL RESULTS

The goal of the study is to develop a generalized model using a dataset (*i.e.*, training set) and the developed model is evaluated with an entirely disjoint dataset (*i.e.*, testing set). The effect of combinational pre-processing techniques is observed to minimize generalized errors. The adopted recursive pre-processing approach has been evaluated using two different scenarios: (1) The dataset $D_1$ is used as a training set using the validation method of split percentage and $D_2$ is applied as a testing set. (2) The dataset $D_2$ is used for training the model using the split percentage method and $D_1$ is used as a testing set. The details about experiments for both scenarios are reported in Table 3 and their results are discussed in the following.

**Table 4  Experimental Results (Scenario 1).**

| | Pre-processing method: | | SR | | |
|---|---|---|---|---|---|
| Training dataset: | | $D_1$ | Testing dataset: | | $D_2$ |
| Split (%) | Algorithms | Accuracy (%) | Precision (%) | Recall (%) | F-measure (%) |
| 100 | DT | 83.27 | 85.13 | 83.11 | 83.52 |
| | LR | 75.25 | 80.19 | 74.40 | 75.68 |
| | SVM | 74.06 | 78.78 | 73.16 | 74.48 |
| | NB | 58.34 | 59.02 | 56.94 | 57.98 |
| | RF | 75.94 | 81.25 | 75.07 | 76.38 |
| 80 | DT | 69.45 | 74.11 | 68.19 | 69.90 |
| | LR | 72.03 | 77.51 | 70.82 | 72.49 |
| | SVM | 71.94 | 77.62 | 70.68 | 72.41 |
| | NB | 57.65 | 58.35 | 56.06 | 57.27 |
| | RF | 75.12 | 80.00 | 74.26 | 75.54 |
| 70 | DT | 72.81 | 78.22 | 71.69 | 73.27 |
| | LR | 72.49 | 78.73 | 71.18 | 72.97 |
| | SVM | 71.06 | 77.51 | 69.56 | 71.56 |
| | NB | 57.00 | 57.35 | 55.86 | 56.69 |
| | RF | 75.25 | 80.19 | 74.40 | 75.68 |
| 60 | DT | 71.47 | 77.34 | 70.14 | 71.96 |
| | LR | 72.44 | 78.83 | 71.10 | 72.93 |
| | SVM | 70.37 | 76.91 | 68.77 | 70.88 |
| | NB | 55.07 | 55.27 | 53.72 | 54.73 |
| | RF | 74.06 | 79.70 | 72.99 | 74.51 |

## Scenario 1

The dataset $D_1$ is used for the training (model trained on tweets) and dataset $D_2$ is used for the purpose of testing. Split percentage is used for the validation because cross-validation method does not support the aim of the study, which uses different datasets for the training and testing purposes. $D_1$ is split into 100%, 80%, 70% and 60% for training as reported in Table 3.

Table 4 shows the performance evaluation of considered machine learning algorithms when recursive pre-processing uses merely the stopword removal (SR) method. It is observed that DT performed with an accuracy of 83.27%, precision of 85.13%, F1 Score of 83.11% and Recall of 83.52% when 100% of dataset $D_1$ is used for training and 100% $D_2$ as testing set. NB performance remained bad amongst the considered algorithms for different experiments. For instance, while a 60% training set is used, naive Bayes produced an accuracy of 55.65%. It is obvious from the results that when more data is used in training, the developed models perform better. It may be noticed that it is not necessary that more cleaning operation leads to a generalised model. Instead, it depends on the pre-processing operations which lead to producing more matching vocabulary for both datasets.

Machine learning algorithms, inherently, tend to behave differently with a different dataset. Besides, a combination of more than one pre-processing technique may also cause different performances in terms of outcomes. It is evident from Fig. 12 that RF performed

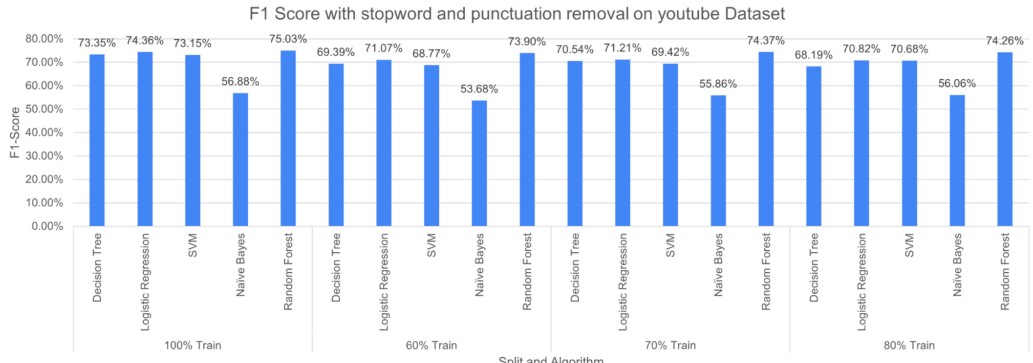

**Figure 12** F-measure with SR + PR ($D_1$ training $D_2$ testing).

better with F-measure when 100% split training is used and achieved the least F-measure at 60% split. NB performance remained bad in comparison with the other four considered algorithms. The NB achieved the highest F-measure of 56.88% (100% split training set) and the least value of 53.68% (60% split training set) is used.

The three pre-processing techniques are combined (*i.e.*, SR + PR + AR) in recursive pre-processing, the results of this experiment are shown in Fig. 13. RF outperformed other considered algorithms 75.19% (at 100% split). The poor performance was again produced by NB with an F-measure of 53.75% (at 60% split).

The experimental results showed that DT outperformed when less number of pre-processing techniques are applied. However, when the number of pre-processing techniques has increased the performance of DT decreases because of its reliance on a number of features which decrease with an increasing number of cleaning operations. The ensemble nature of RF produced better results with increasing the pre-processing operations. NB produced poor results for the experiments, the possible reason is its nature of probabilities. The dataset is unstructured data *i.e.*, text data, thus, the probabilistic approach may not perform well in these situations.

## Scenario 2

This scenario uses dataset $D_2$ as training and dataset $D_1$ for testing. Table 5 shows the results of this scenario when SR and PR pre-processing operations are performed. RF outperformed in this scenario also with accuracy, precision, recall and F1 score of 74.54%, 76.10%, 74.08% and 74.41% respectively at 100% split of training data. Naive Bayes performed very poorly with almost 60% of accuracy around all the splits (*i.e.*, 100%, 80%, 70% and 60%). The increase in pre-processing operations for RF tends to perform better than DT.

Figure 14 shows F1 score for the considered algorithms at different splits when a single pre-processing operation SR (stopword removal) is applied. The outperforming algorithm, in this case, is DT with F1 score of 73.12% at 100% split. It is observed that the value of F1 score decreases with the decrease in training data. NB performed poorly with the F1 score

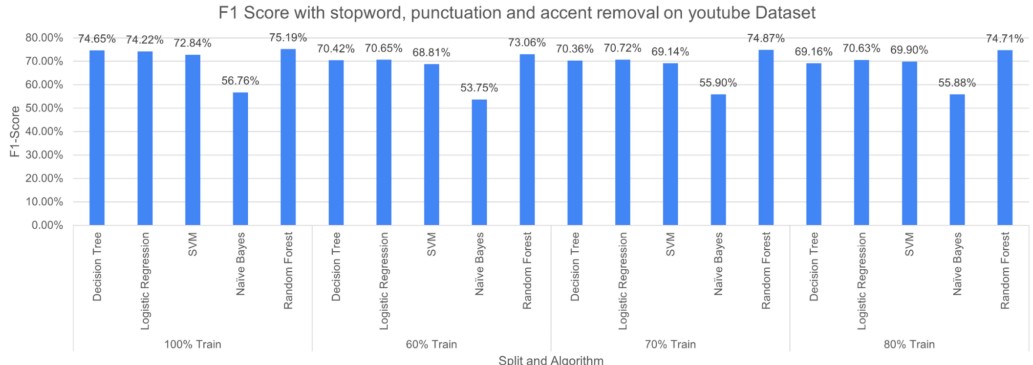

**Figure 13** F-measure with SR + PR + AR ( $D_1$ training $D_2$ testing).

**Table 5** Experimental results (Scenario 2).

| Pre-processing method: | | | SR + PR | | |
|---|---|---|---|---|---|
| Training dataset: | | $D_2$ | Testing dataset: | | $D_1$ |
| Split (%) | Algorithms | Accuracy (%) | Precision (%) | Recall (%) | F-measure (%) |
| 100 | DT | 72.92 | 74.17 | 72.49 | 72.79 |
| | LR | 70.33 | 71.05 | 70.01 | 70.23 |
| | SVM | 70.75 | 72.04 | 70.23 | 70.61 |
| | NB | 59.13 | 62.09 | 56.82 | 59.38 |
| | RF | 74.54 | 76.10 | 74.08 | 74.41 |
| 80 | DT | 70.71 | 71.21 | 70.48 | 70.62 |
| | LR | 70.21 | 70.70 | 69.97 | 70.12 |
| | SVM | 70.54 | 71.54 | 70.11 | 7042 |
| | NB | 59.50 | 61.91 | 57.66 | 59.73 |
| | RF | 71.00 | 71.87 | 70.63 | 70.89 |
| 70 | DT | 70.71 | 71.23 | 70.48 | 70.62 |
| | LR | 69.33 | 69.81 | 69.08 | 69.24 |
| | SVM | 69.96 | 70.85 | 69.55 | 69.84 |
| | NB | 59.79 | 62.84 | 57.59 | 60.05 |
| | RF | 69.96 | 70.91 | 69.53 | 69.84 |
| 60 | DT | 70.17 | 71.12 | 69.74 | 70.05 |
| | LR | 68.63 | 69.01 | 68.40 | 68.54 |
| | SVM | 69.75 | 70.81 | 69.27 | 69.62 |
| | NB | 61.25 | 62.85 | 60.22 | 61.43 |
| | RF | 66.79 | 68.73 | 65.76 | 66.61 |

of 57.55% at 100% split. The difference in F-measure scores varies around 2% to 3% for NB at different splits.

Figure 15 reports three pre-processing techniques (SR, PR and AR) are applied together; the performance of DT decreases from 73.12% to 72.75% and RF increases from 71.50% to

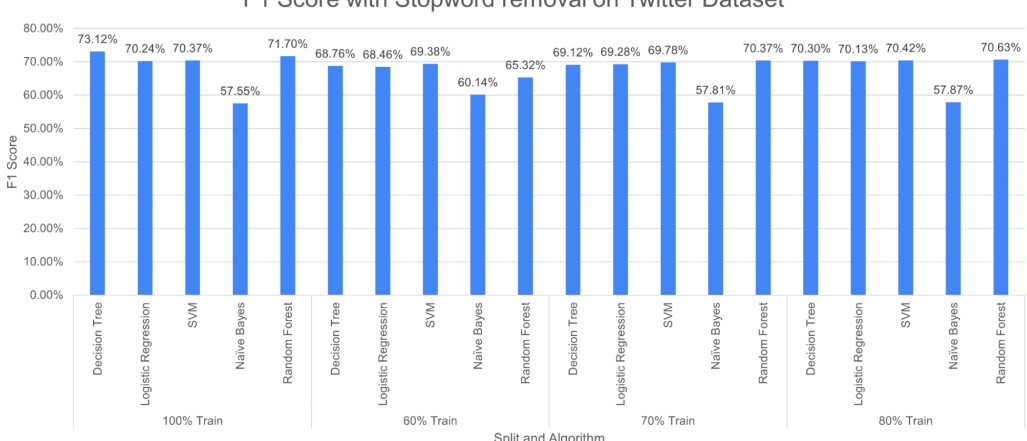

**Figure 14** F1 score with SR + PR with $D_1$ as testing.

72.75% at 100% split. The performance of the model decreases with a decrease in training percentage. The very poor-performing model throughout the study is naive bayes.

## Comparison with baseline

We compared models created using suggested pre-processing techniques from the experiments in scenarios 1 and scenario 2. We trained classification models based on datasets $D_1$ and $D_2$ separately with a stratified split of 15%. The comparative results using dataset $D_1$ with existing studies, our ML models outperformed the XGBOOST (eXtreme Gradient Boosting) and LGBM (Light Gradient Boosting Machine) models. However, the proposed combinatorial pre-processing techniques with ML algorithms could not achieve better results than that of transfer learning models which use pre-processing as well as neural networks, such as mBERT (multilingual Bidirectional Encoder Representations from Transformers) and dehatebert-mono-arabic. The experimental results, in which dataset $D_2$ is applied, show that the ML models outperformed all the models reported in *Akhter et al. (2020)* for both scenario 1 in section 'Scenario 1' and scenario 2 in section 'Scenario 2'. These results suggest that an objective approach in selecting pre-processing steps can improve the performance of a simple ML model compared to classical and boosting techniques.

## Answering the research questions (RQs)

The experimental results provide the answers to the research questions (RQs). To answer **RQ1**, the results in scenario 1 in section 'Scenario 1' and scenario 2 in section 'Scenario 2' clearly suggest that pre-processing techniques improve the performance of the classification model based on cross-platform data. However, the performances of models depend upon pre-processing approaches and their combinations. The answer to **RQ2**, it can be observed from the results that in scenario 1 in section 'Scenario 1' where the $D_1$ is used for training, the most effective pre-processing technique is SR, because data in tweets is limited by characters and also the language used in tweets is substandard. Thus, other pre-processing

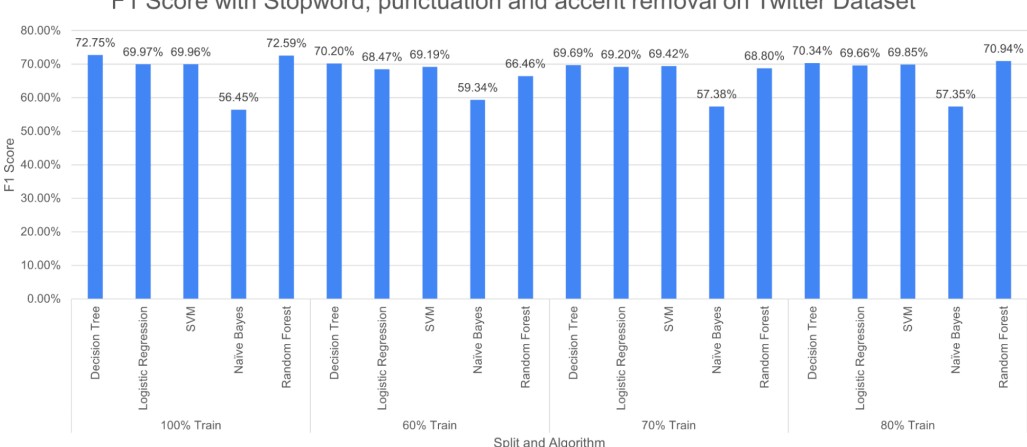

**Figure 15** F1 score with SR + PR + AR with $D_1$ as testing.

**Table 6** Comparison of model performances with existing literature.

| Camparison with Existing Literature | | | |
|---|---|---|---|
| **Dataset 1** | | **Dataset 2** | |
| **Model** | **F-1 Measure** | **Model** | **F-1 Measure** |
| dehatebert-mono-arabic *Das, Banerjee & Saha (2021)* | 0.88062 | LR | 0.972 |
| mBERT *Das, Banerjee & Saha (2021)* | 0.84 | RF | 0.961 |
| SVM | 0.829 | SVC | 0.959 |
| LR | 0.826 | SimpleLogistic | 0.959 |
| RF | 0.822 | LogitBoost *Akhter et al. (2020)* | 0.949 |
| XGBoost *Das, Banerjee & Saha (2021)* | 0.76072 | DT *Akhter et al. (2020)* | 0.949 |
| LGBM *Das, Banerjee & Saha (2021)* | 0.76667 | SVM *Akhter et al. (2020)* | 95.5 |

techniques would not affect it. In scenario 2 in section 'Scenario 1', the combination of SR and PR performed well because the language used in the comment is standard and has no limit of characters. AR requires the use of sophisticated language, which is usually unavailable on social media platforms. Table 6 reports the answer to **RQ3**, that is, pre-processing affects significantly over classical machine learning models. However, pre-processing techniques do not make a significant impact on NLP models such as BERT. Thus, it is evident from experimental results that pre-processing techniques can improve the performance of the classification model at a certain level.

This study focused on developing a model using one dataset and evaluating the model using entirely another dataset. Therefore, the comparison of the results of scenario 1 (details in section 'Scenario 1') and scenario 2 (details in section 'Scenario 2') to the existing literature would be biased. However, experiments have been performed to compare the developed model by applying combinational pre-processing techniques using dataset-1 ($d_1$) as well as dataset-2 ($d_2$) separately. These experiments used the $d_1$ for the training as well as testing in order to compare the results with the existing literature. Similarly,

experiments were performed using $d_2$ using the proposed approach of developing the model. These experimental results are reported in Table 6 and their confusion matrix for the developed model using the proposed approach is presented in Fig. 16.

The confusion matrices reported in Fig. 16 show the effectiveness of the proposed approach in which combinational pre-processing techniques are used to develop the model. It is observed that using the proposed approach to develop the model outperformed the models available in the existing literature (see details in Table 6). Thus, it is inferred from experimental results shown in section 'Scenario 1', section 'Scenario 2', and Table 6 the proposed model that adopts combinational pre-processing techniques provides worth mentioning results. The effectiveness of the proposed approach is observed when the same dataset is used for the training as well as testing (see Table 6). Besides, it also provided significant results when one dataset is used for training and an entirely disjoint dataset is used for the testing (see section 'Scenario 1' and section 'Scenario 2').

## DISCUSSION

The experimental results showed DT performed better amongst other considered algorithms with an accuracy of 83.72%. The increase in pre-processing operations at the training model decreases the performance of DT. However, the RF performed vice-versa than the DT. NB performed poorly throughout the experiments because of its inability to generate a generalized model. The experiments have been carried out using different split percentages such as 100%, 80%, 70% and 60%.

The experiments in which $D_1$ is used as the training set and SR pre-processing operation is applied produced better results. Likewise, the experiments, which considered $D_2$ as training and SR + PR, both pre-processing operations applied, yield better results.

The difference in performance when the different dataset is used in training is due to the different nature of the content available in both datasets ($D_1$ and $D_2$). $D_2$ dataset contains purely comment text while $D_1$ also has commentary, which is why when the model is trained with $D_1$, it can generalize better as compared to the model trained with $D_2$. The results from scenario 1 and 2 can't be used for comparison with existing work as the dataset for training and evaluation vary from the dataset in existing literature.

The use of machine learning algorithms to detect offensive terms in the text can raise ethical considerations, particularly in the context of privacy, data protection, and cultural norms. Machine learning models can be biased because of the data being used as the training set. Therefore, it should be ensured that the data, being used for training the models, is representative of the target population and does not perpetuate any biases or stereotypes. Furthermore, the models need to be transparent and explainable, allowing users to understand the models' working mechanisms and the reasons behind their decisions. In addition, it should be noted that offensive language detection requires access to users' text data, which can be sensitive and personal. Thus, data collection and processing be ensured in compliance with data protection regulations and their intended purposes. The datasets used in this study are publicly available, hence, data privacy and protection were ensured.

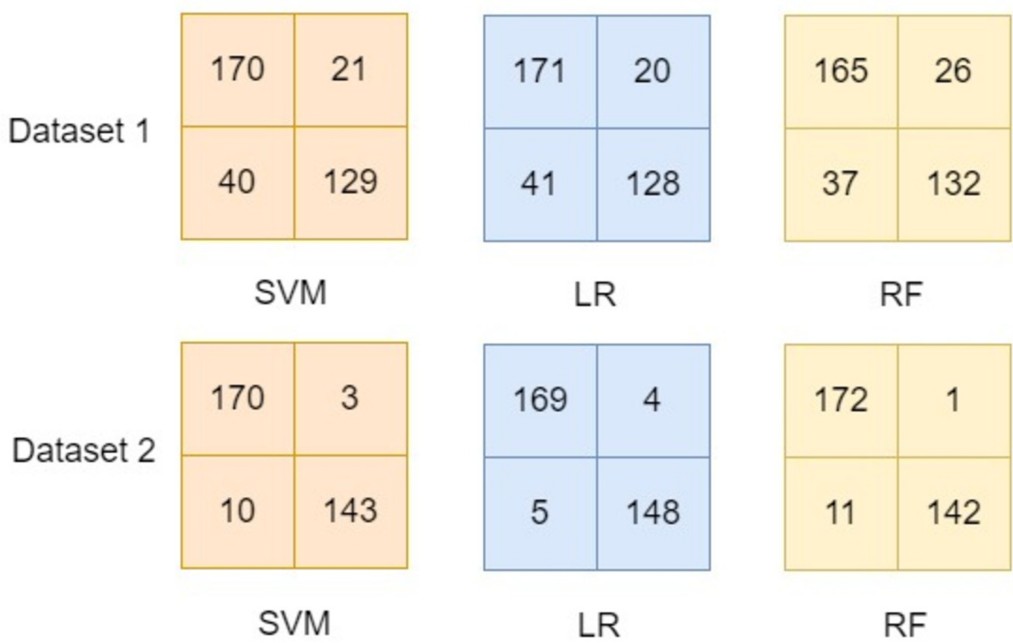

**Figure 16 Confusion matrix for the models in comparison.**

# CONCLUSIONS AND FUTURE WORK

This study deals with offensive language detection in Urdu, a resource-poor language resulting in a challenging task for classification in the field of NLP. The significant contribution of the study is building several ML classification models for Urdu using different pre-processing approaches and cross-platform datasets. This study proposes a combinatorial pre-processing approach that uses different pre-processing operations stand-alone and combined to build an effective classification model. Five machine learning algorithms have been applied, namely, DT, LR, SVM, NB, and RF, which are trained using a dataset and tested on an entirely different dataset. The main limitations of the machine learning algorithm rise due to the bias in a dataset, thus, resulting in higher generalized error. Therefore, to avoid such errors, mutually exclusive datasets are used for experiments in this study.

This study shows the potential of using cross-platform datasets with different pre-processing approaches to evaluate models to create better generalization of ML models in offensive language detection. Also, the models created in this study are compared with benchmarks. The results show that the technique provides better results, hence proving the pre-processing impacts model performance in terms of cross-platform data, but the combination of pre-processing techniques may vary from dataset to dataset.

The main limitation of this study is pre-processing operations. Different languages may have different pre-processing operations in comparison to Urdu. It is, therefore, expected that performing all operations applied in this study to other low-resource languages may not provide similar outcomes.

For future research, this study highlights the need for further exploration of low-resource languages and their specific challenges in abusive term detection. This study may be considered as an attempt towards investigating offensive term detection for low-resource languages. Novel machine-learning techniques may be developed that can improve performance. These systems can help identify and remove offensive content and reduce the burden on human moderators.

Additionally, modeling may be performed over deep learning models such as RNN, LSTM and BILSTM and transferring learning models such as BERT. More pre-processing approaches will also be considered, which is challenging for low-resource language.

### Funding
The authors received funding from the Deanship of Scientific Research at Najran University for this research through a grant (NU/RG/SERC/12/33) under the Research Groups Funding program at Najran University, Kingdom of Saudi Arabia. The funders had no role in study design, data collection and analysis, decision to publish, or preparation of the manuscript.

### Grant Disclosures
The following grant information was disclosed by the authors:
The Deanship of Scientific Research at Najran University: NU/RG/SERC/12/33.
The Research Groups Funding program at Najran University, Kingdom of Saudi Arabia.

### Competing Interests
The authors declare that there are no competing interests.

### Author Contributions
- Muhammad Owais Raza conceived and designed the experiments, analyzed the data, authored or reviewed drafts of the article, and approved the final draft.
- Naeem Ahmed Mahoto conceived and designed the experiments, analyzed the data, authored or reviewed drafts of the article, and approved the final draft.
- Mohammed Hamdi performed the experiments, performed the computation work, prepared figures and/or tables, and approved the final draft.
- Mana Saleh Al Reshan performed the experiments, performed the computation work, prepared figures and/or tables, and approved the final draft.
- Adel Rajab performed the experiments, performed the computation work, prepared figures and/or tables, and approved the final draft.
- Asadullah Shaikh conceived and designed the experiments, analyzed the data, authored or reviewed drafts of the article, and approved the final draft.

### Data Deposition
Data is available at GitHub:

https://github.com/owais4321/Resource-Poor-Language-Urdu-Dataset

The D1 dataset is available at GitHub: https://github.com/MaazAmjad/Urdu-abusive-detection-FIRE2021 (https://arxiv.org/abs/2207.06710).

Authors: Maaz Amjada, Alisa Zhilab, Grigori Sidorova, Andrey Labunetsc, Sabur Butta, Hamza Imam Amjadd, Oxana Vitmana and Alexander Gelbukh

Institutions: Instituto Politécnico Nacional (IPN), Center for Computing Research (CIC), Mexico

Ronin Institute for Independent Scholarship, United States

Independent Researcher, United States,

Moscow Institute of Physics and Technology, Russia

The D2 dataset is available at GitHub: https://github.com/pervezbcs/Urdu-Abusive-Dataset

Authors: MUHAMMAD PERVEZ AKHTER, ZHENG JIANGBIN, IRFAN RAZA NAQVI, MOHAMMED ABDEL MAJEED AND MUHAMMAD TARIQ SADIQ

Institutions School of Software and Microelectronics, Northwestern Polytechnical University, Xian, China,

School of Computer Science and Technology, Northwestern Polytechnical University, Xian, China

School of Automation, Northwestern Polytechnical University, Xian, China.

## Supplemental Information

Supplemental information for this article can be found online at http://dx.doi.org/10.7717/peerj-cs.1524#supplemental-information.

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
