# Peer review of "Detection of offensive terms in resource-poor language using machine learning algorithms"

_PeerJ Computer Science, doi:10.7717/peerj-cs.1524_

## Round 0.1 · original submission · Major Revisions

The authors should address the reviewers comments clearly.

Reviewer 1 has requested that you cite specific references. You may add them if you believe they are especially relevant. However, I do not expect you to include these citations, and if you do not include them, this will not influence my decision.

Reviewer 1 ·

Basic reporting

Dear Authors,
I have reviewed the manuscript “Detection of offensive terms in resource-poor language using machine learning algorithms” Manuscript ID: 80376v1 that has been submitted for publication in the: PeerJ Computer Science, and I have identified a series of aspects that in my opinion must be addressed in order to bring a benefit to the manuscript.
The article under review will be improved if the authors address the following aspects in the text of the manuscript:
1. It is best to write numerical results in the abstract to better confirm and present the results as many accuracy and validation measures.
2. The authors should describe the proposal in detail in figure 2.

Experimental design

3. In “Table 4. Experimental Results (Scenario - 1)” and Table 5, I think the results can be improved because the accuracy values are somewhat low and can be improved by using an optimization algorithm with a classifier.
4. Using deep learning as features extraction and then a classifier will give better results than the algorithms used and will also reduce the time.

Validity of the findings

5. A more accurate evaluation measure can be used as the MCC.

Additional comments

6. Updating the references with the new research for the years 2021 and 2022 can be used
10.21608/mjeer.2021.146277
10.1109/NRSC49500.2020.9235095

·

Basic reporting

The paper is interesting and deals with a novel topic and I recommend acceptance with major revision.
I have provided notes below. Critical points are in red

The is a problem within text referencing, I would have liked to see more current references.
I have provided an example in line 39, this paper needs to be copy edited.
I have a problem with the lack of discussion (4.3 which is buried in the results) I think this should be a clear section 5 on it own and incorporate 4.4. while the question ansewers are sound, these should also be contextualised with the literature a little more


Introduction
Reads well but the structure of 56-74 is a bit weak, this should be reorginaised
The contributions should be outlined in the abstract developed in the discussion and explicit in the conclusion they seem out of place here.
Grammatical
24 Was it necessary to capitalise the ICT
35 -36 amend repetition (add supporting citation):
“Particularly, social media platforms have emerged as 36 sources of viable channels for people to spread their views. The social media including Facebook, Twitter 37 and other such platforms have offered for discourse and raise viewpoints about numerous topics and 38 stories round the world”
39 grammar and needs citation. The use of maybe here is problematic, the authors should be definitive. Suggest
“cyberbullying and online harassment maybe is on the rise due … (Citation, 2022)“
48 What literature? add citation
49 Do you have evidence of the little effort?

56 Start a new paragraph at ….. This study

Methods
How were the word clouds constricted which programs what were the parameters? Were nodes defined, was NVivo used, etc etc etc
130 reference formatting error, this formatting error is found throughout the paper

Comparison with existing literature
281 Why is the comparison with the literature (ie discussion) is short and needs to be expanded I am not convinced that there are ONLY 2 papers the authors could have engaged with to expand this part.

Experimental design

The paper is interesting and deals with a novel topic and I recommend acceptance with major revision.
I have provided notes below. Critical points are in red

The is a problem within text referencing, I would have liked to see more current references.
I have provided an example in line 39, this paper needs to be copy edited.
I have a problem with the lack of discussion (4.3 which is buried in the results) I think this should be a clear section 5 on it own and incorporate 4.4. while the question ansewers are sound, these should also be contextualised with the literature a little more


Introduction
Reads well but the structure of 56-74 is a bit weak, this should be reorginaised
The contributions should be outlined in the abstract developed in the discussion and explicit in the conclusion they seem out of place here.
Grammatical
24 Was it necessary to capitalise the ICT
35 -36 amend repetition (add supporting citation):
“Particularly, social media platforms have emerged as 36 source of viable channels for people to spread their views. The social media including Facebook, Twitter 37 and other such platforms have offered for discourse and raise viewpoints about numerous topics and 38 stories round the world”
39 grammer and needs citation. The use of may be here is problematic, the authors should be definitive. Suggest
“cyberbullying and online harassment may be is on rise due … (Citation, 2022)“
48 What literature? add citation
49 Do you have evidence of the little effort?

56 Start a new paragraph at ….. This study

Methods
How were the word clouds constricted which programs what were the parameters. Were nodes defined, was NVivo used, etc etc etc
130 reference formatting error, this formatting error is found throughout the paper

Comparison with existing literature
281 Why is the comparison with the literature (ie discussion) is short and need to be expanded I am not convinced that there are ONLY 2 papers the authors could have engaged with to expand this part.

Validity of the findings

.

Additional comments

.

---

## Round 0.2 · Major Revisions

Please clarify the following questions in your research clearly:

1- What are the resource-poor languages used in the study, and what are the specific challenges of detecting offensive terms in these languages?

2- What machine learning algorithms were used in the study, and how were they selected? How do these algorithms compare to other methods for detecting offensive terms in text?

3- What features or characteristics of the language were used to train the machine learning algorithms? How do these features impact the accuracy of the algorithms?

4-How was the accuracy of the machine learning algorithms measured and validated? What were the false positive and false negative rates, and how do these rates impact the algorithms' usefulness in practice?

5- What are the ethical considerations of using machine learning algorithms to detect offensive terms in text? How was privacy and data protection ensured in the study?

6- What are the implications for future research and practical applications?

7- What are the limitations of the study, and how might they impact the generalizability and applicability of the findings to other resource-poor languages or contexts?

8- What are the potential biases or limitations of using machine learning algorithms to detect offensive terms in text, and how might these biases impact the outcomes of the study?

---

## Round 0.3 · Major Revisions

According to the results received in Scenario 1 (DT1), F1 of your ML models didn't have advantages over the current models, whereas, F1 of your ML Models achieved in Scenario (2) only very low significance compared with the current model. Please clarify further justification to accept the paper.

Also there is no confusion matrix as I asked you before to show it in your revised paper. Please try to make further comparison between your work and the other current studies based on the confusion matrix.

---

## Round 0.4 · accepted · Accept

I am happy to inform you that the reviewers are satisfied with the modifications of the revised manuscript.

Reviewer 1 ·

Basic reporting

Clear and unambiguous

Experimental design

Research question well defined, relevant & meaningful. It is stated how research fills an identified knowledge gap.

Validity of the findings

All underlying data have been provided; they are robust, statistically sound, & controlled.